# The effects of microbiome-targeted therapy on cognitive impairment and postoperative cognitive dysfunction — A systematic review

**Saiko Sugita**[1], **Peggy Tahir**[2], **Sakura Kinjo**[3]*

**1** Department of Anesthesiology, Nippon Medical School, Tama-Nagayama Hospital, Tokyo, Japan,
**2** University of California San Francisco Library, University of California, San Francisco, San Francisco,
California, United States of America, **3** Department of Anesthesiology and Perioperative Care, University of
California, San Francisco, San Francisco, California, United States of America

* Sakura.Kinjo@ucsf.edu

## Abstract

### Background

The gut-brain axis involves bidirectional communication between the gut-microbiota and central nervous system. This study aimed to investigate whether probiotics and/or prebiotics, known as Microbiome-targeted Therapies (MTTs), improve cognition and prevent postoperative cognitive dysfunction (POCD).

### Methods

Relevant animal and human studies were identified using a systematic database search (PubMed, EMBASE, Cochrane Library, and Web of Science), focusing on the effects of MTTs on inflammation, perioperative and non-perioperative cognitive impairment. Screening and data extraction were conducted by two independent reviewers. The Risk of bias was assessed using the SYRCLE's risk of bias tool for animal studies. The revised Cochrane risk of bias tool (RoB 2) was used for human studies.

### Results

A total of 24 articles were selected; 16 of these involved animal studies, and 8 described studies in humans. In these papers, the use of MTTs consistently resulted in decreased inflammation in perioperative and non-perioperative settings. Out of 16 animal studies, 5 studies (2 associated with delirium and 3 studies related to POCD) were conducted in a perioperative setting. MTTs improved perioperative cognitive behavior and reduced inflammation in all 5 animal studies. Eleven animal studies were conducted in a non-perioperative setting. In all of these studies, MTTs showed improvement in learning and memory function. MTTs showed a positive effect on levels of pro-inflammatory cytokines and biomarkers related to cognitive function. Among the 8 human studies, only one study examined the effects of perioperative MTTs on cognitive function. This study showed a reduced incidence of POCD along with improved cognitive function. Of the remaining 7 studies, 6 suggested that MTTs improved behavioral test results and cognition in non-perioperative

**Data Availability Statement:** All relevant data are within the paper and its Supporting information files.

**Funding:** The author(s) received no specific funding for this work.

**Competing interests:** The authors have declared that no competing interests exist.

environments. One study failed to show any significant differences in memory, biomarkers of inflammation, or oxidative factors.

## Conclusion

In the studies we examined, most showed that MTTs decrease inflammation by down-regulating inflammatory cytokines and oxidative stress in both perioperative and non-perioperative settings. In general, MTTs also seem to have a positive effect on cognition through neural, immune, endocrine, and metabolic pathways. However, these effects have not yet resulted in a consensus regarding preventative strategies or treatments. Based on these current research results, MTTs could be a potential new preventative strategy for cognitive impairment after surgery.

## Introduction

The concept of the gut-brain axis, the bidirectional communication between gut microbiota in the gastrointestinal tract and brain, has recently been confirmed by a growing number of studies [1, 2]. The microbiome has been implicated as having an impact on host function well beyond the gut, including obesity, diabetes, cardiovascular disease, autism, behavior, and motor activity [3–5], along with neurocognitive disorders including mild cognitive impairment (MCI) and Alzheimer's disease (AD) [2, 6–9]. In addition, the link between gut microbiota and Postoperative cognitive dysfunction (POCD) has been getting attention [10]. POCD is a cognitive change or decline after surgery with anesthesia, often persisting for weeks or months. The occurrence of POCD has been estimated to be between 25 and 42% on postoperative day 7 or discharge, and 10% at three months after surgery in patients aged at least 60 years old [11, 12]. Such prolonged deficits in attention, memory, and concentration can lead to a higher risk of permanent cognitive impairment or dementia [11, 13]. This data, together with the fact that the world's population over 60 years old is projected to increase to 33% by the year 2050 [14], raises the importance of identifying preventive strategies for cognitive impairment.

A possible explanation for the link between gut microbiota and cognition could be pro-inflammatory cytokines and oxidative stress in the central nervous system (CNS). Recent studies have demonstrated that neuroinflammation is a hallmark of AD and POCD in both human and animal models [11, 12]. This neuroinflammation is thought to be brought on by either disruption of the blood-brain barrier (BBB), activation of microglia and astrocytes, and/or oxidative stress induced by surgery and anesthesia [13, 15]. Therefore, various clinical interventions have been used to reduce neuroinflammation. However, none of these have been universally adopted as a standard of care [12, 15]. Recent preclinical studies have shown that some perioperative drugs such as anesthetics, opioid analgesics, and antibiotics could possibly affect postoperative cognitive function by altering the composition or diversity of gut-microbiota [10, 12, 16, 17]. And surgical procedures (e.g., intestinal resection) themselves could change the balance of gut microbiota [18]. There is a growing body of evidence that perioperative stressors (e.g., emotional, environmental, physiological, surgical insult, medications, and infections) lead to gut microflora changes and dysbiosis. The composition of the gut microbiota may undergo rapid and often extreme changes and potentially cause multiple organ dysfunction (e.g., neurological, respiratory, gastrointestinal, cardiovascular, and renal) [19]. Literature

suggests that microbiome-targeted therapies (MTTs), especially supplementation with probiotics and /or prebiotics, improve the balance of gut-microbiota and modulate neurological function via immune, metabolic and endocrine pathways [20, 21]. Thus far, most of the MTT-related perioperative studies are focused on the infection (e.g., ventilator-associated pneumonia and surgical site infection) [22], and much less literature has focused on cognitive function as an outcome.

Therefore, this systematic review aims to examine current evidence for the effects of MTTs on cognitive impairment, including POCD. In addition, we will discuss the possible pathophysiologic mechanisms of cognitive impairment by neuroinflammation via the gut-brain axis.

## Methods

### Eligibility criteria

Randomized trials or cross-over studies of animals or humans (elderly, at least adult) investigating the effects of MTTs on cognition were considered. Studies with outcomes other than cognition or not written in English were excluded. The study protocol for human studies was registered with the International prospective register of systematic reviews (PROSPERO). (ID: CRD42020178197) The study protocol for animal studies was not registered with PROSPERO.

### Literature search

This systematic review was conducted according to the guidelines of the Preferred Reporting Items for Systematic Reviews and Meta-Analyses (PRISMA) checklist (S1 File). Two authors (S.S., P.T.) separately searched for publications using PubMed, EMBASE, Cochrane, and Web of Science for relevant literature (from their inception to October 2021) using the following terms: microbiome, microbiota, gastrointestinal microbiome, probiotic, probiotics, cognitive dysfunction, delirium, cognitive impairment, confusion, mental deterioration, cognition disorders, anesthesia/adverse effects, postoperative complications/therapy, postoperative complications/prevention and control, dysbiosis/therapy, inflammation/drug therapy, inflammation/complications, aged, elderly and geriatric. Additional citations were sought using reference lists of relevant articles and gray literature.

### Data selection and extraction

We screened all the titles and abstracts and removed less relevant articles according to criteria including (P) population: aged or at least adult, (I) intervention: MTTs such as supplementation of probiotics and/or prebiotics or fecal microbiome transplantation compared with placebo, (O) outcomes: pre-specified clinical outcomes such as cognitive improvement or deterioration, postoperative complications, and differences in biomarkers, expression of protein or mRNA. All publications included in the study are written in English.

### Data collection and assessment of quality of papers

Data were collected in both animal and human studies. In animal studies, the following information was included: animal model, age, type of probiotics or prebiotics, duration of interventions, and outcomes. In clinical studies, age, disease, types of probiotics, and duration of interventions and outcomes.

Two authors (S.S., S.K.) independently assessed the quality of clinical studies using the "Revised Cochrane Risk of Bias tool for randomized trials (RoB2) [23], which includes five categories; 1) randomization process, 2) deviation from intended interventions, 3) missing

outcome data, 4) measurement of the outcome and 5) selection of the reported result. We assessed the quality of each domain answering the signaling questions listed in the guidance. Based on the results of these five categories, we determined the overall risk of bias for each trial.

In addition, the quality of animal studies was assessed using SYRCLE's risk of bias tool [24] which includes 1) selection bias, 2) performance bias, 3) detection bias, 4) attrition bias, and 5) reporting bias, and 6) other bias. This tool is based on the Cochrane RoB tool and has been adjusted for aspects of bias that play a specific role in animal intervention studies. It is the most recommended tool for assessing the methodological quality of animal interventional studies [25].

## Results

### Literature search

The results of the literature search were divided into two groups: 1) MMTs and cognitive dysfunction in animal studies, 2) MTTs and cognitive dysfunction in clinical studies. A total of 1631 articles were identified (1616 by the database search and 15 by manual search). 1458 papers were excluded for lack of relevance. Another 29 articles were removed since their abstracts or full text were not available online or not written in English. The final set of studies included 16 articles on animal studies and 8 articles on human studies (Fig 1). Out of 16 animal studies, 5 studies were associated with perioperative cognitive dysfunction (2 for delirium, 3 for POCD). The eight human studies were all randomized, double-blind, placebo-controlled trials. Seven of these were conducted in non-perioperative settings, and one was in a perioperative setting. In these human studies, the reported age range was 50 to 100 years old.

### Risk of bias

S1-S5 Tables in S5 File show the revised Cochrane risk of bias tool (RoB 2) in each category; 1) randomization process, 2) deviation from the intended interventions, 3) missing outcome data, 4) judgment in measurement of the outcome, and 5) judgment of selection of reported results for the clinical studies. S6 Table in S5 File shows the summary of S1-S5 Tables in S5 File. No study was completely free of risk of bias. Three studies [26–28] were classified as"low" in overall risk of bias. Four studies [29–32] were classified as "some concerns", and one study [33] was classified as "some concerns to high." For animal studies, an overview of SYRCLE's risk of bias assessment is presented in S7 Table in S5 File. Fifteen studies were graded for 6 types of bias. Except for one study which showed "low risk" in all categories, [34] all studies included "unclear" in at least one of the domains due to lack of information.

### Gut-microbiota and MTTs, and cognitive function in animals

**Animal studies investigating perioperative cognitive changes.** Out of 16 animal studies that assessed the efficacy of MTTs on cognitive function, 5 studies investigated perioperative cognitive changes in animals. Four of them are interventional studies using prebiotics or probiotics [17, 35–37] (Table 1), and the other focused on fecal microbiota transplant [38]. Common findings among these studies were improved memory function and/or behavior assessed by behavioral tests, including maze test, novel object recognition test, and open field test [17, 35–38]. The differences in the gut microbiota composition such as α- and β- diversity between intervention and control rodents were also consistent in these studies [16, 35, 36, 38]. These studies detected the altered composition of gut microbiota after anesthesia/surgery. According to Jiang et al., quantitative real-time polymerase chain reaction (qRT-PCR) of fecal samples

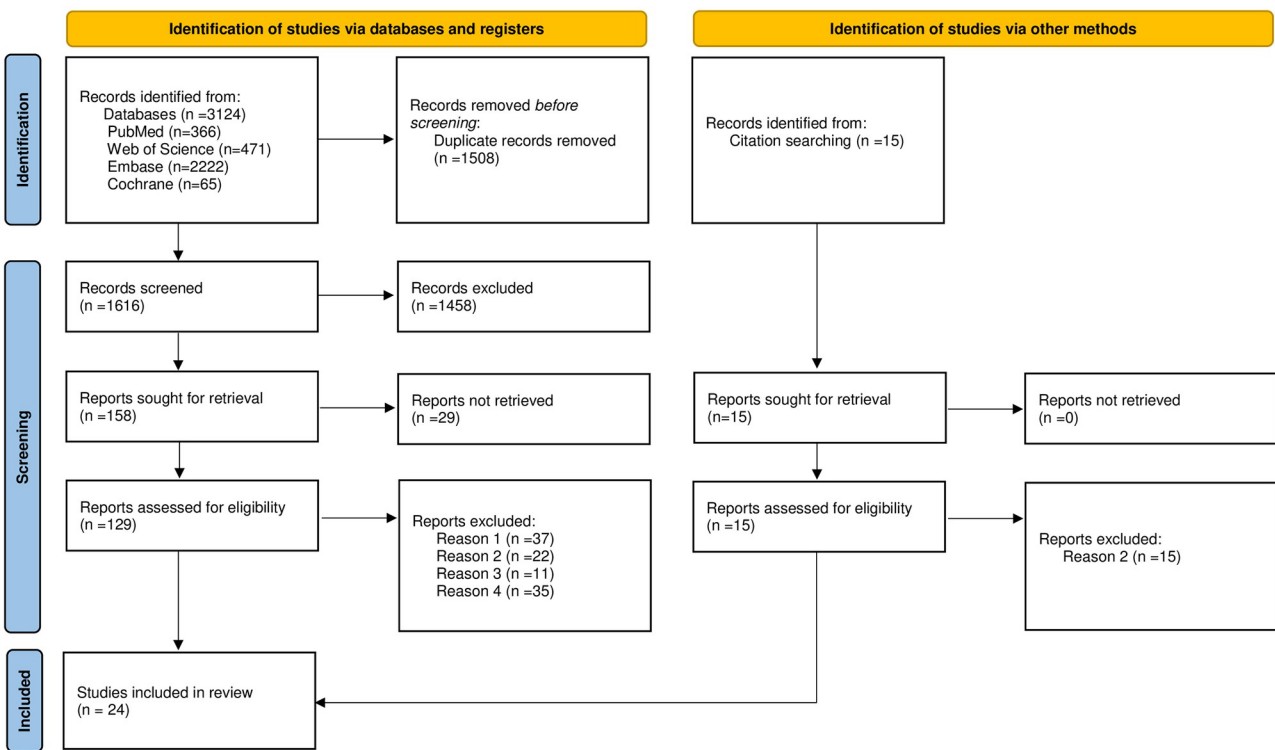

**Fig 1. Preferred reporting items for systematic reviews and meta-analyses (PRISMA) flow diagram.** * Reason 1; Perioperative studies, but focused on other than cognitive function. Reason 2; Reviews or Not interventional studies. Reason 3; Populations are not of interest. Reason 4; Others (Details are available in Supplemental Information).

obtained 48 hours after surgery in mice pretreated with VSL#3 (a mixture of several probiotic cultures) for 10 days before surgery detected that 8 out of 37 types of bacteria that were altered by anesthesia/surgery had returned to baseline [36]. This recovery in these 8 types of bacteria was not found in the group of VSL#3 without surgery. They also reported these changes in gut microbiota were correlated with deficits in reference memory. Yang et al. reported that 3 weeks of prebiotic treatment in advance of anesthesia/surgery significantly alleviated cognitive decline observed after surgery, and changed the beta diversity of the gut microbiome [35]. Liufu et al. also showed that anesthesia/surgery induced significant alterations in the components of gut microbiota in older mice. And the delirium-like behavior observed in these mice after anesthesia/surgery was mitigated by preoperative treatment with *Lactobacillus (L.) salivarius and L. rhamnomus* [17]. In another interventional study of probiotics conducted by Wen et al., the applying a Lactobacillus mix also reduced anesthesia/surgery-induced POCD in aged mice [37]. The remaining research associated with POD investigated the efficacy of fecal transplants on cognitive function [38]. In addition to a significant difference in the diversity of gut microbiota between POD and non-POD mice, they also reported fecal bacteria transplants from non-POD mice improved the abnormal postoperative behavior in the pseudo-germ-free mice. In contrast, the fecal transplants from POCD mice showed no significant effect. Another study that assessed the differences in POCD between mice with different fecal bacteria was conducted by Zhan et al. This study reported that the Dehalobacteriaceae family and the Dehalobacterium genus were potentially crucial for the diagnosis of POCD, and that these bacteria were significantly correlated with the results of their MWMT [16].

**Table 1. Summary of perioperative interventional studies with pre- or probiotics in rodents.**

| First Author, Year | Animal Model, Age | Pre-/Probiotic Agent, Duration | Main Results (compared with model group) |
|---|---|---|---|
| Yang, 2018 | Sprague-Dawley rats 8 months old | Galacto-oligosaccharide perioerative 3 weeks | Improved poestoperative cognitive performance in NOR test |
| | | | Reduced Iba-1 positive cell in the hippocampus in immunohistochemical staining |
| | | | Decreased of M1 phenotype microglia and suppressed microglial overactivation |
| | | | Downregulated level of protein expression of IL-6 in hippocampus |
| | | | Changes in microbial community towards potentially anti-inflammatory status |
| Jiang, 2019 | POCD model 18 months old | VSL#3 (probiotic blend with 8 bacterial strains) preoperative 10 days and postoperative 7 days | Preventative effects in reference memory in MWMT impaired by ansthesia/surgery |
| | | | Altered beta diversity and differed 37 genera were revealed by 16S rRNA sequence |
| | | | Potentially critical 8 types of microbial speices for the impaied reference memory were detected by 16SrRNA sequence and qRT-PCR |
| Liufu, 2020 | Anesthesia/surgery model 9 and 18 months old | *Lactobacillus rhamnosus* GG preoperative 20 days | Anethesia/surgery induced age-dependent behavioral change in BFT, OFT, and Y-maze test |
| | | | Probiotics attenuated delirium-like behavioral change induced by anethesia/surgery |
| | | | Decreased level of proinflammatory cytokines in brain |
| | | | Reduction of mitochondrial dysfunction in the hippocampus induced anesthesia/surgery |
| Wen, 2020 | Anesthesia/surgery model 6 weeks and 18 months old | *Lactobacillus casei* CICC 6108 *Lactobacillus rhamonsus* GG *Lactobacillus helveticus* ND01 Sodium butyrate 4 weeks | Improved spatial memory learning ability in Y-maze test reduced after anesthesia/surgery |
| | | | Increased expression of tight junction proteins between endothelial cells in the hippocampus |
| | | | Decreased the blood-brain barrier permeability |

Abbreviations used: POCD, postoperative cognitive dysfunction; MWMT, morris water maze test; OFT, open field test; NOR, novel object recognition; BFT, bried food test

Overall, these findings suggest that perioperative MTT therapies could lead to behavioral improvements by modulating gut microbiota composition.

**Animal studies investigating non-perioperative cognitive changes.** Here we summarize 11 non-perioperative interventional studies. Six studies used rodents in an AD model or drug-induced neuroinflammation model, 3 used SAMP8 mice, and 2 used dementia model mice [2, 16, 34, 39–41] (Table 2). A previous study reported that rodents with cognitive decline have a significantly different composition of their gut microbiota compared with age-matched normal mice [42]. Chen et al. reported prebiotic supplementation improved the diversity of gut microbiota in AD model mice. Further, all of the studies with AD model mice succeeded in showing improvement in spatial learning and memory performance in hippocampus-dependent behavioral tests. Acetylcholine (ACh) and acetylcholine esterase (AChE) are the pathological targets of AD and cognitive dysfunction, including delirium. Some interventional studies showed increased ACh and decreased AchE.

On the other hand, the expression of dopamine (DA), serotonin (5-HT), and BDNF are generally believed to be associated with memory deficiencies [40]. Between studies with SAMP8 and those using dementia model mice, common findings were improvement in spatial learning and memory function, and increased levels of DA, 5-HT, and BDNF in serum and brain. In addition, elevated levels of SCFAs were commonly detected across models. SCFAs

**Table 2. Summary of interventional studies with pre- or probiotics in rodents with cognitive decline.**

| First Author, Year | Animal Model, Age | Pre-/Probiotic Agent, Duration | Main Results (compared with model group) |
|---|---|---|---|
| Liu, 2015 | Vascular dementia model 6 weeks old | *Clostridium butyricum* 6 weeks | Improvement in spatial learning ability in MWMT |
| | | | Increased diversity of intestinal bacteria in PCR-DGGE profiles |
| | | | Increased level of SCFAs in the feces (p<0.01) and in the brain (p<0.05) |
| | | | Increased the protein level of BDNF (p<0.01) |
| Musa, 2016 | LPS-induced neuroinflammation model 8 weeks old | *Lactobacillus fermentum* LAB9 or *Lactobacillus casei* LABPC 4 weeks | Improved spatial learning and memory in MWMT (p<0.001) |
| | | | Reduction of acetylechorineesterase activity in brain tissue (p<0.001) |
| | | | Increased level of antioxidants (p<0.001) in brain tissue |
| | | | Reduced level of proinflammatory cutokines (p<0.01) in brain tissue |
| Kobayashi, 2017 | Alzheimer's disease model 10 weeks old | *Bifidobacterium breve* strain A1 11 days | Amelioration of cognitive dysfunction in working memory and long-term memory in the Y maze test and passive avoidance test (p<0.05) |
| | | | Elevated plasma level of acetate (p<0.05) |
| | | | Suppressing effect on the hippocampal expression of inflammation and immune-reactive genes |
| Bonfilli, 2017 | Alzheimer's disease model (3xTg-AD) 8 weeks old | SLAB51 (probiotic blend with 9 live bacterial strains) 16 weeks | Improved hippocampus-dependent recognition memory in NOR test (p<0.05) |
| | | | Increased fecal content of SCFAs (p<0.05) and reduced plasma concentrations of pro-inflammatory cytokines (p<0.05) |
| | | | Increased plasma concentraions of gut hormones with neuroprotective effect (p<0.05) |
| Nimgampalle, 2017 | Alzheimer's disease model 12 weeks old | *Lactobacillus plantarum* MTCC 1325 60 days | Shortened escape latency time in MWMT (p<0.05) |
| | | | Maintained healthy neurons with prominent nuclei inhistopathological examination |
| | | | Increased the level of acetylchorine (p<0.05) and decreased level of acetylchorineesterase (p<0.05) in hippocampus and cerebral cortex |
| Chen, 2017 | Alzheimer's disease model 10 months old | Fructo-oligosaccharides from *Morinda officinalis* 4 weeks | Ameliorated the learning and memory dysfunction in MWMT |
| | | | Maintained superior cell morphology in HE staining of the small intestine |
| | | | Recovered the deficient indexes of the diversity of gut microbiota |
| | | | Decreased the level of serum proinflammatory cytokines |
| | | | Changes in antioxidative molecules in brain |
| | | | Promoted secretion of neuroprotective neruotransmitters |
| Huang, 2018 | Senescence Accelated Mouse Prone 8 16 weeks old | *Lactobacillus paracasei* PS23 12 weeks | Decreased anxiety-like behavior in OFT (p<0.05) and memory impairment in MWMT (p<0.05) |
| | | | Increased level of dopamine, serotonin in striatum and hippocampus (p<0.05) |
| | | | Higher levels of BDNF and anti-inflammatory cytokinesin the serum (p<0.05) |
| | | | HIgher levels of antioxidative enzymes in the serum and in the hippocampus (p<0.05) |
| Corpuz, 2018 | Senescence Accelated Mouse Prone 8 14 weeks old | *Lactobacillus paracasei* K71 43 weeks | Improved spatial learning and memory in Barns test (p<0.05) and fear-motivated learning and short-term memory in Y-maze test (p<0.05) |
| | | | Increased level of serotonin in the serum and brain (p<0.05) |
| | | | Increased expression of *Bdnf* mRNA and BDNF protein in the hippocampus (p<0.05) |
| Chunchai, 2018 | Dietary-induced dementia model 13 weeks old | Xyo-oligosaccharide *Lactobacillus casei* 12 weeks | Attenuated gut dysbiosis by decreasing F/B ratio |
| | | | Reversed hippocampal dysplasticity in fEPSP slope of LTP |
| | | | Reduced ROS production, mitchondrial depolarization and swelling in brain |
| | | | Preserved microglial morphology paramerters |
| | | | Attenuated impairment of learning and memory in MWMT |

*(Continued)*

**Table 2.** (Continued)

| First Author, Year | Animal Model, Age | Pre-/Probiotic Agent, Duration | Main Results (compared with model group) |
|---|---|---|---|
| Wang, 2019 | Alzheimer's disease model 9 months old | GV-971 (mixture of oligosaccharides) 3 months | Enhanced spatial learning and memory performance in MWMT |
| | | | FMT from GV-971-treated mice resulted in decreased Th1 cells in the brain of recipient mice injected with aggregated Aβ |
| | | | Decreased brain Th1 cells, microglial activation, and brain cytokines level |
| | | | Reduced concentration of phenylalanine and isoleucine in the feces |
| | | | Attenuated Th1-related neuroinflammation by the mechanisms listed above |
| Lin, 2021 | Senescence Accelated Mouse Prone 8 3 months old | *Lactobacillus plantarum* GKM3 14 weeks | Increased lomg-term memory in the passive avoidance test (p<0.05) and learning memory in the active avoidance test (p<0.05) |
| | | | Reduced level of oxidative stress in brain (p<0.05) |
| | | | Less accumulated amyloid-β in brainin immunohistochemical examination (p<0.05) |
| | | | Maintained arrangement of neurons, cell structure, and morphology in the hippocampus |

Abbreviations used: SCFA, short-chain fatty acids; BDNF, brain-derived neurotrophic factor; MWMT, morris water maze test; NOR, novel objective recognition; IL-, interleukin; FMT, fecal microbiota transplantation; Th1, Type1 helper T cell

might affect the brain via direct humoral effects, endocrine and immune pathways, and neural routes [21]. Therefore, these findings suggest that MTTs may positively impact cognitive function by regulating neurotransmitters and SCFAs.

## Gut-microbiota and MTTs on cognitive function in humans

**Human studies investigating perioperative cognitive changes.** To the best of our knowledge, one RCT conducted by Wang et al. [31] is the first and only clinical study examining the effect of perioperative MTTs on cognitive function in humans. These were elderly patients who underwent non-cardiac surgery. In this study, the patients were assigned to take oral probiotics (a combination of *Bifidobacterium longum*, *Lactobacillus acidophilus*, and *Enterococcus faecalis*) or a placebo from hospital admission until discharge. The incidence of POCD was lower in the probiotic group than in the placebo group (5.1% vs. 16.4%, P = 0.046). In addition, their levels of plasma IL-6 and cortisol after surgery were lower, compared with the control group (IL-6: -117.90 ± 49.15 vs. -14.93 ± 15.21, P = 0.044; cortisol: -158.70 ± 53.52 vs. 40.98 ± 72.48, P = 0.010).

**Human studies investigating non-perioperative cognitive changes.** Over the past decade, some studies have suggested that MTTs may affect neurological disorders. However, these have looked primarily at psychiatric disorders such as anxiety, mood disorders, and depression [2, 43]. Yet, some studies and systematic reviews have recently reported the association between gut-microbiota or MTTs and cognitive frailty [44, 45]. Most of these associations include effects on MCI, AD, Parkinson's disease, and dementia (Table 3). An analysis of gut microbiota composition in AD patients showed that their microbiota had lower levels of bacterial strains with anti-inflammatory properties and a higher abundance of strains with pro-inflammatory properties [46, 47].

Further, this alteration was associated with a shift in biomarkers of systemic inflammation toward a pro-inflammatory state [47]. Yet, some studies implied that such changes could be reversed through probiotic and prebiotic intake, or specific dietary changes, including a modified Mediterranean ketogenic diet [48] and lipid-rich milk [49]. An Interventional study for patients with AD who received a milk drink containing *L. Lactobacillus*, *L.*

**Table 3. Summary of interventional clinical trials on cognitive function in populations with cognitive disorders.**

| First Author, Year, Study Design | Population | Age(y), N | Interventional Agent, Duration | Main Results (compared with control group) |
|---|---|---|---|---|
| Akbari, 2016 Randomized, Double-blind, Placebo-controlled | Patients with Alzheimer's disease | 60–95 n = 60 | Probiotic milf containing *Lactobacillus acidphlius, Lactobacillus casei, Lactobacillus fermentum, Bifidobacterium bifidum* 12 weeks | Approximately 30% greater improvement in MMSE score |
| | | | | Approximately 50% greater reduction of serum hs-CRP |
| | | | | Approximately 25% greater reduction of serum MDA |
| | | | | Improvement in insulin metabolism and lipid metabolism |
| Kobayashi, 2019 Randomized, Double-blind, Placebo-controlled | Elderly subjects withmemory complaints | 50–80 n = 117 | *Bifidobacterium breve* A1 12 weeks | Significant improvement in the subscale of 'immediate memory' in RBANS and total score in MMSE in the subjects with low RBANS total score at baseline |
| | | | | Significant difference in total MMSE score and in the subscale of 'recall' in RBANS in the high-score group |
| | | | | No significant differences in blood parameters |
| Hwang, 2019 Randomized, Double-blind, Placebo-controlled | Individuals with Mild Cognitive Impairment | 55–85 n = 100 | *Lactobacillus plantarum* C29-fermented soybean (DW2009) 12 weeks | Greter improvemet rate in the combined cognitive function (z = 2.36), especially |
| | | | | in the attention domain (z = 2.34) in the computerized neurocognitive function tests |
| | | | | Positive association between serum BDNF levels and the change of combined cognitive function |
| | | | | No significant differences in vital signs, boday mass index, and laboratory profiles |
| Tamtaji, 2019 Randomized, Double-blind, Placebo-controlled | Patients with Alzheimer's disease | 55–100 n = 79 | *Lactobacillus acidphlius, Bifidobacterium bifidum, Bifidobacterium longum,* elenium 12 weeks | Approximately 20% greater improvement rate in MMSE score |
| | | | | Increased level of antioxdant molecules and decreased level of hs-CPR in blood |
| | | | | No significant differences in the level of oxdative molecules |
| | | | | Improvement in insulin metabolism and lipid profiles |
| Agashi, 2019 Randomized, Double-blind, Placebo-controlled | Patients with Alzheimer's disease | 65–90 n = 60 | *Lactobacillus fermentum, Lactobacillus plantarum, Lactobacillus acidophilus, Bifidobacterium longum, Bifidobacterium lactis, Bifidobacterium bifidum* 12 weeks | Change rate between the scores in Test Your Memory test at the onset and the offset of the trial did not reach the statistical difference |
| | | | | No statisticaly significant affects on the level of inflammatory cytokines,antidxdant, or oxdative molecules |
| Xiao, 2020 Randomized, Double-blind, Placebo-controlled | Patients with Mild Cognitive Impairment | 50–79 n = 80 | *Bifidobacterium breve* A1 16 weeks | Approximately 30% greater improvemet rate in RBANS total score |
| | | | | Improvement in the domain of immediate memory, visuospatial/constructional, and delayed memory |
| | | | | Approximately 5% greater change rate in JMCIS score |
| Sanborn, 2020 Randomized, double-blind, placebo-controlled | Community-dwelling adults including ones with cognitive impairment | 52–75 n = 145 | *Lactobacillus rhamnosus* GG 12 weeks | Significantly greater improvement in the NIH Toolbox Total Cognition Score compared with persons without cognitive impaiment |
| *Wang, 2021 Randomized, double-blind, placebo-controlled | Non-cardiac planned surgical patients | 60–90 n = 120 | *Bifidobacterium longum, Lactobacillus acidophilus, Enterococcus faecalis* From admission until discharge (more than 7 days) | Approximately 70% lower incidence of postoperative cognitive impairment |
| | | | | Greater improvement in MMSE score |
| | | | | Decreased plasma level of proinflammatory cytokine and cortisol |
| | | | | No significant differences in postoperative pain, sleep quality, and gastrointesinal function recovery |

Abbreviations used: MMSE, Mini-mental status examination; RBEANS, Repeatable Battery for Assessment of Neuropsycological Status, JMCIS; Japanese version of the MCI Screen test

hs-CRP, high-sensitivity C-reactive protein; MDA, malondialdehyde; BDNF, brain-derived neurotrophic factor

* perioperative study

*casei*, *B. bifidum*, and *L. fermentum* for 12 weeks demonstrated improvement in the mini-mental state examination (MMSE) score, high sensitivity (hs-) CRP, and malondialdehyde (MDA), a product of oxidative stress [29]. Another study with AD patients that assessed the effects of combined use of selenium and probiotic agents including *L. acidophilus*, *B. bifidum*, *and B. longum* also increased MMSE scores [30]. In this study, patients taking selenium and probiotics had decreased levels of serum hs-CRP, insulin, and homeostasis model of assessment-insulin resistance (HOMA-IR). Dysregulation of glucose metabolism and insulin resistance have previously been reported to be linked to the pathogenesis and progress of AD [50]. On the other hand, a similar study of AD patients that examined changes in cognitive function and biochemical factors failed to detect positive effects of probiotic agents containing either Lactobacillus or Bifidobacterium [33]. The authors pointed out the severity of AD and its irreversibility in the loss of synapses and progression of neuro-frailty as a reason that might explain why the probiotics did not have any effect. Meanwhile, an RCT showed that *L. plantarum* C29-fermented soybean intake in individuals with MCI for 12 weeks improved cognitive function, especially with attention. These improvements were positively correlated with increased levels of serum BDNF [27]. Another study of 117 elderly subjects with memory deficits who took supplements of the probiotic *B. breve* for 12 weeks revealed a significant improvement in their subscale "immediate memory", based on neuropsychological testing and MMSE [26]. Microorganisms, especially *Bifidobacterium* are known to have the capacity of producing propionate and modulating proper functioning of the hypothalamic-pituitary-adrenal axis (HPA), which is essential for cognitive processes such as learning and memory [20]. A study by Mohammadi et al., using probiotic yogurt and multispecies probiotic capsule supplementation for 6 weeks did not affect the HPA axis. However, mental health parameters, including a general health questionnaire (GHQ) and depression anxiety and stress scale (DASS) were significantly improved in the intervention group [51].

## Discussion

### Main findings

We investigated whether MTTs improve cognition in perioperative and non-perioperative settings. We identified 16 animal studies. They showed that MTTs had favorable effects on cognition in both perioperative and non-perioperative settings.

Among 8 human studies, only one study examined the effects of MTTs in perioperative settings. This study showed that the incidence of POCD was lower in the probiotic group than the placebo group. Six out of 7 non-perioperative studies showed improvement of cognition with the use of MTTs.

### Epidemiology of probiotics

According to studies that assessed the administration of probiotics by physicians, 51% of medical doctors had advised probiotics to at least some of their patients in their practice [52]. One study showed that a growing number of inpatients received probiotics as part of their care in U.S hospitals [53]. In this study, the use of probiotics increased from 1.0% of discharged patients in 2006 to 2.9% of discharged patients in 2012.

### Risks and benefits of probiotics

Table 4 shows a list of some commercially available products. There is a wide array of probiotic products, and the effectiveness and safety of particular products are often not objectively

**Table 4. Summary of risks, benefits, and examples of commercial products for major microbial genes.**

| Genus / Related Neurotransmitter | Spieces/Strains | Risk | Benefits | Examples of commercial products |
|---|---|---|---|---|
| *Lactobacillus* /GABA, Acetylecholine | *L. casei* | Infections | Priventive effects of Influenza | Cheese, Yogurt, Oat, Barley |
| | | including sepsis | Decreased frequency of constipation | Yakult fermented dairy drink |
| | | Mild gas | Decreased risk of bladder cancer | Danone® Actmel |
| | *L. paracasei* | Infections | Peduced allergy | Yakult fermented dairy drink |
| | | | Relief of skin sensivity | |
| | *L. acidophilus* | | Increase in H.pylori eradication rate | Greek yogurt/ Cheese |
| | | | Decrease in antibiotics-related diarrhea | Kefir/ Sauerkraut |
| | | | Relief of irritable bowel syndrome | Miso/ Tempeh |
| | *L. rhamnosus* | | Attenuated seasonal allergy | Dietary supplements |
| | | | Improved vaginal health | ATCC 7469 (Use for research purposes only) |
| | *L. gasseri* | | Decreased H.pylori | |
| *Streptcoccuss* / Serotonin *Prevotella* | *S. Thermophilus* | Infections | Reduced antibiotics-related diarrhea | Yogurt |
| | | | | Cheese |
| | | | | Dietary supplements |
| | *P. copri* | Associated with | Associated with glucose tolerance, insulin resistance | |
| | | rheumatoid arthritis | | |
| *Bacillus* / Noradrenaline, Dopamine | *B. Coagulans* | | Enhanced Immune system | Muesli,Cereal bars |
| | | | Improved vaginal health | Kimuchi |
| | | | | Kombucha |
| *Bifidobacterium* / GABA | *B.longum* | | Prevention of carcinogenesis | Breast milk, Yogurt, Cheese |
| | | | | Mushrooms, Artichoke, Broccoli, |
| | | | | Beetroot, and Seaweed |
| | *B. breve* | | Reduced symptoms of inflammatory bowel disease | |
| | *B. bifidum* | | Reduced risk of infection from food borne pathogens | |
| *Saccharomyses* / Noradrenaline | *S. boulardii* | | Prevention in C. difficile-related diarrhea | Dietary Supplement |

measured. In general, probiotics and prebiotics are thought by many to have health-promoting effects. However, a review paper published in 2019 reported that in some cases, adverse effects of probiotics had been reported. These include systemic infections, gastrointestinal side effects, skin complications, and endocarditis [54]. Most frequently reported is fungemia, caused by *Saccharomyces cerevisiae* and its subspecies, *S.boulardii* [55–60]. *S. cerevisiae* (baker's yeast) is a common colonizer of the human gastrointestinal system as a benign organism. It is used in Europe to treat and prevent *C.difficile-* associated diarrhea. Also, *Bifidobacterium* and *Lactobacillus* have been reported as pathogenic germs [61–67]. All reported fungemia or septicemia were detected in immunosuppressed patients, critically ill patients, and elderly patients. It should be noted that probiotics should be used cautiously in such patients. Further, trimethylamine N-oxide (TMAO), a metabolite of intestinal flora, has been shown to contribute to the pathogenesis of many diseases, including cardiovascular disease and AD [68, 69]. High circulating TMAO can aggravate postoperative hippocampal-dependent cognitive dysfunction through increased pro-inflammatory cytokines, microglial activation, and reactive oxygen species in aged rodents.

## Neuroinflammation, aging, and POCD

It is well established epidemiologically, that aging, pre-existing cerebrovascular disease, alcohol intake, opioid use, and low educational level are associated with POCD [12]. The activation of the HPA and cholinergic anti-inflammatory pathway (in response to surgical stress) are also potential perioperative factors [13, 15]. Recently, neuroinflammation is thought to be one of the important contributive mechanisms related to apoptosis and decreased synaptic plasticity and synthesis of neurons. Aging is associated with changes that induce a chronic low-grade pro-inflammatory environment, referred to as "inflammaging". Inflammaging can result in disruption of the brain blood barrier (BBB), activation of microglia and astrocytes, and the transformation of microglial phenotypes from a resting state (M2) towards an inflammatory phenotype (M1), called microglial priming. Such alternation is dubbed 'neruroinflammaging'. In addition to this underlying neuroinflammaging, surgery may cause additional systemic inflammation, resulting in damage to endothelial and perivascular cells by increasing the synthesis of TNF alpha via the activation of NFkB, and reducing tight junction proteins. All together, these effects aggravate the permeability of BBB [13]. Further, microglial priming likely contributes to the development of perioperative neuroinflammation, as microglial priming in the elderly tends to delay once stimulated by perioperative injury, producing cytokines, reactive oxygen species, and other pro-inflammatory modulators. Danger-associated molecular patterns (DAMPs), such as the high-mobility group box 1 protein (HMGB-1) and S100 calcium binding protein (S100beta) are also reported to be involved in the development of postoperative cognitive deficits. Both surgery and anesthesia could drive the release of HMGB1 from dendritic cells and macrophages depending on the severity of tissue injury [13]. In parallel with activated microglia and astrocytes, a reduction of brain-derived neurotrophic factor (BDNF) is also observed following surgery, along with alternations in neurogenesis, synaptogenesis, and neural plasticity. All these mechanisms could induce neuroinflammation after surgery in the areas crucial for cognitive function, such as hippocampus and striatum, leading to the development of POCD.

## Microbiome-targeted therapy (MTT)

MTTs, especially supplementation using probiotic and prebiotic agents, have become of more interest in the last decade. According to the World Health Organization, probiotics are defined as 'live microorganisms which when administrated in adequate amounts, confer a health benefit on the host', whereas prebiotic is "a selectively fermented ingredient that results in specific changes in the composition and/or activity of the gastrointestinal microbiota, thus conferring benefit(s) upon host health". Synbiotic refers to preparations where probiotics and prebiotics are combined. Microorganisms can survive gastric acid pH and bile in the gastrointestinal tract by adhering to the intestinal mucosa [70]. MTTs can be classified mainly into supplementation using probiotics and/or prebiotics; and fecal microbiota transplantation [71–75]. In this discussion, we summarize the former. In the literature, probiotic and/or prebiotic supplementation has generally been intended to: (1) enhance gastrointestinal barrier function, (2) improve immunity to certain infectious bacteria, (3) improve colonization resistance, (4) reduce inflammation. These may contribute to reduction in gut pH, production of antimicrobial substances, agglutination of harmful bacteria, increasing of gut mucus secretion and intestinal protective substances such as short-chain fatty acids (SCFA) [76].

## Benefits of MTTs on inflammation

Aging is associated with an overall chronic low-grade pro-inflammatory environment, termed "inflammaging" [77]. Age-related changes in the microbiome reduce the beneficial effects of

gut microbiota due to decreased diversity and inflammaging, and contribute to the breakdown of the intestinal barrier, loss of bacterial containment, and chronic activation of the host immune system [78]. MTTs are expected to prevent bacterial translocation and pro-inflammatory modulators from circulating systemically, based on the mechanisms described in the previous section. In animal models with AD, dementia, or senescence-accelerated mouse prone 8 (SAMP8), both perioperative and non-perioperative interventional studies using probiotics and prebiotics consistently showed decreased levels of pro-inflammatory cytokines, microglial activation, and oxidative status, including interleukin 6 (IL-6) and TNF-$\alpha$ (Tables 1, 2). There are a number of studies providing evidence that MTT reduces inflammation in humans. For example, a meta-analysis examined the effect of prebiotics and synbiotics on systemic inflammation and showed that prebiotics and synbiotic supplementation in populations with systemic disease was associated with decreased inflammatory markers, including CRP, IL-6, and TNF-$\alpha$ [79]. Other studies showed that in a perioperative setting, the use of probiotics in planned surgical patients showed the downregulation of markers such as IL-6, IL-1$\beta$, and CRP [80–82].

## MTTs and POCD

While the number of studies in this area has dramatically increased over the past ten years, it is not yet possible to accurately generalize or make conclusions about the effects of MTTs on cognitive impairment, including POCD. Given the clinical studies thus far using probiotics or prebiotics, MTTs can improve cognitive function in populations with cognitive decline such as AD and MCI. Cognitive impairment in AD or MCI usually progresses over time. POCD however, progresses over a shorter period. In this regard, one should be cautious in extrapolating these results to POCD.

In addition, while there is consensus that POCD refers to a broad spectrum of clinical conditions characterized by acute and persistent POCD, speed of processing, and executive functioning, POCD is not clearly defined by either the Diagnostic and Statistical Manual of Mental Disorders, International Classification of Diseases, or biomarkers. Numerous neurocognitive tests (e.g., Rey Auditory Verbal Learning Test, Trail Making Test, Grooved Pegboard Test, Digit Span Test) have been used to assess different brain functions in the literature [83]. This heterogenicity adds more complexity to conducting research and drawing conclusions. Thus far, there is only one RCT using probiotics in humans in the literature. The ambiguous definition of POCD and the difficulties in scientifically comparing pre-and postoperative cognitive function in elderly patients could hinder the large-scale clinical adoption of probiotics and prebiotics.

## Gaps in reported data, and recommendations

To better understand the impact of MTTs on cognition, it would be helpful to include the following data in future clinical studies:

1. Duration and timing of MTTs: The current literature does not have sufficient data on the effective duration of MTTs on cognition. In the non-operative setting,12 weeks of MTTs in humans is commonly used. In the perioperative settings, probiotics were used from hospital admission to discharge in the study by Wang et al. Timing of MMTs, whether before surgery or after surgery, needs to be further explored.

2. Types and dose of MTTs: Thus far, *Lactobacillus and Bifidobacterium* species are commonly studied in both animal and clinical studies and have shown promise. Different types and combinations of strains and dosage levels need to be studied further.

3. Assessment of baseline cognition: Assessment of patients' baseline cognitive function should be evaluated before the administration of MTTs. Ideally, different domains of neurological function (e.g., memory, executive function) should be assessed. In addition, cognitive assessment should be performed in acute or sub-acute phases and months after surgery.

## Limitations

There are limitations to this study. First, we excluded some articles because their abstracts or full text were unavailable online. However, it is possible that some of these articles have data relevant to this study. In addition, it is possible that our search terms did not capture all related articles. Second, several articles reporting the positive effects of MTTs here are obtained from animal studies. Nagpal et al. reported on host species-specific signatures of the gut microbiome in rodents and their similarities/differences from humans [84]. They showed that the mice microbiota appears closer to humans than rats based on β-diversity. They also demonstrated a higher Firmicutes–Bacteroidetes ratio in humans than in rodents. The human microbiota is dominated by Bacteroides, while the mouse gut is predominated by members of the family S24-7 and rats have a higher abundance of Prevotella. Also, fecal levels of lactate are higher in rodents than humans, while acetate is highest in human feces [84]. Given these differences between species, one should be cautious in applying the results to humans.

## Final remarks

Still, given the reported evidence thus far, probiotic, prebiotic, and synbiotic therapies can improve the composition of gut microbiota and gut permeability that typically increases with aging. A synthetic graphical overview of possible pathways in the microbiome-gut-brain axis in POCD is presented in Fig 2. MTTs may improve surgery-induced inflammation by increasing the production of SCFAs. Further, microorganisms enhance the synthesis of beneficial neurotransmitters such as BDNF in brain regions crucial for learning and memory function, which could ameliorate bacterial translocation, systemic inflammation, or neuroinflammation. In this way, it may be possible for MTTs to reduce the severity of POCD; however, the evidence supporting MTTs is still premature. Therefore, further basic and clinical research, especially larger randomized controlled studies using different microbiome strains, dosage levels and their combinations. Also, studies using duration of administration as a variable with specific probiotic and prebiotic formulas, correlated to measurable biological and clinical outcomes relevant to cognitive impairment would allow more objective conclusions to be made.

Probiotics and other microorganisms that have reached the intestinal tract are taken up by M cells (membranous cells) in the upper layer of Peyer's patches and captured by dendritic cells in the lower layer of Peyer's patches. Information within the microbial antigens is recognized by toll-like receptors expressed on dendritic cells and transmitted to T cells. Probiotics may affect immune function through activation of T cells, proliferation of intestinal epithelial cells, promotion of IgA production, and suppression of inflammation. Treg, one of the major T cells in the intestinal tract along with Th17, secrete inhibitory cytokines in a gut-dependent manner. Meanwhile, SCFAs fermented from prebiotics are recognized by FFA2 expressed on immune cells. These SCFAs then reach the brain. SCFAs promote anti-inflammatory effects through the inhibition of histone deacetylases (HDACs) and the upregulation of IL-10. Further, some gut microbiota have the capacity of producing beneficial neurotransmitters and proteins such as serotonin, GABA, acetylcholine and BDNF. Thereby, supplementation with

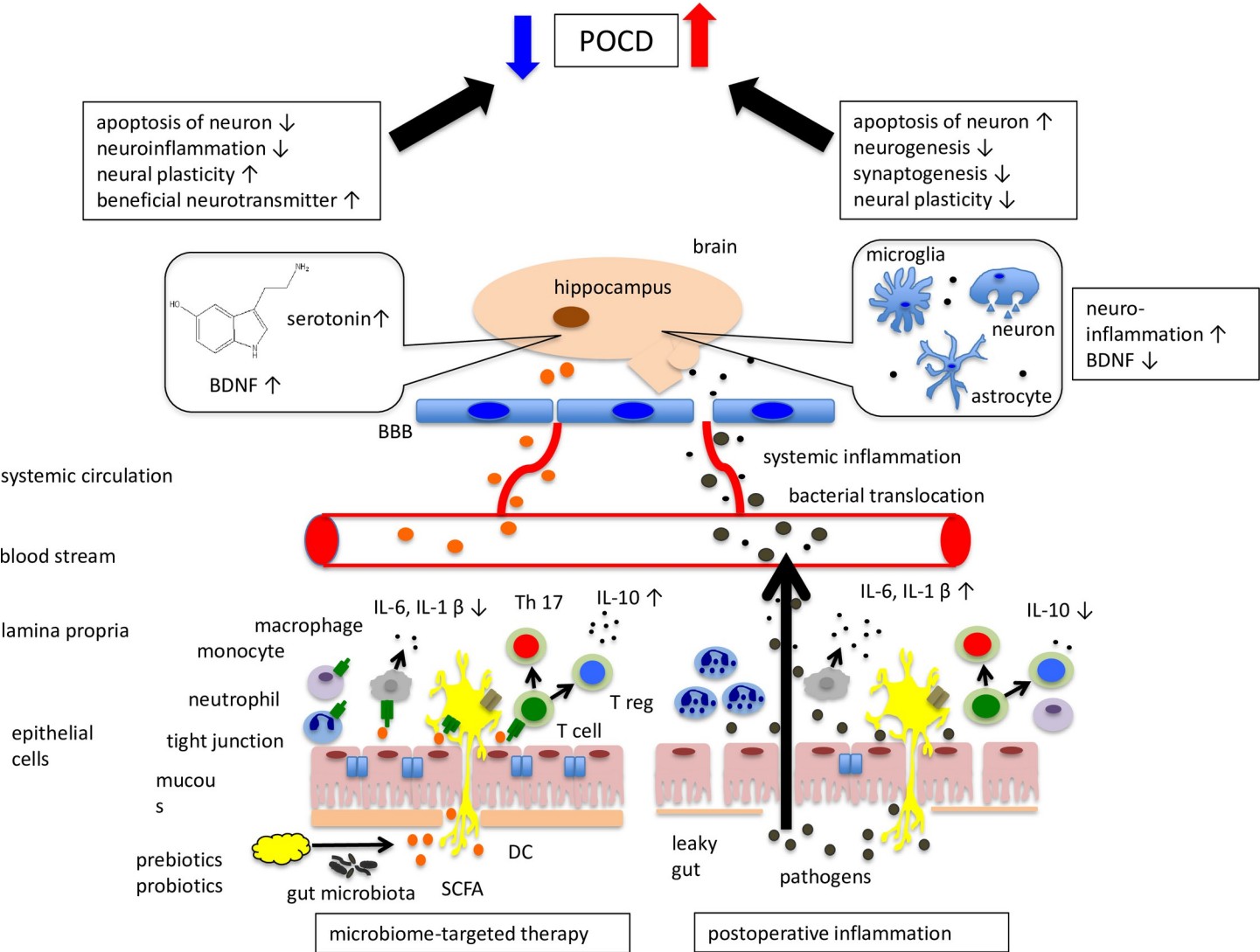

**Fig 2. A synthetic graphical overview of potential pathways in the microbiome-gut-brain axis in POCD.**

probiotics and prebiotics may lead to neuroprotective effects and a reduction in apoptosis of neurons induced by an inflammatory response due to surgical trauma.

## Supporting information

**S1 File. PRISMA checklist.**
(PDF)

**S2 File. Search strategies appendix.**
(PDF)

**S3 File. Protocol synopsis.**
(PDF)

**S4 File. Excluded studies.**
(PDF)

**S5 File. SYRACLE's risk of bias tool for the interventional studies of probiotics and prebiotics and Cochrane risk of bias tool for the clinical interventional studies.**
(PPTX)

## Author Contributions

**Conceptualization:** Saiko Sugita.

**Data curation:** Saiko Sugita, Peggy Tahir.

**Formal analysis:** Saiko Sugita.

**Methodology:** Saiko Sugita.

**Software:** Saiko Sugita.

**Supervision:** Sakura Kinjo.

**Writing – original draft:** Saiko Sugita.

**Writing – review & editing:** Peggy Tahir, Sakura Kinjo.

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
