## [Decision Letter · Decision Letter 0]

2 Feb 2022

PONE-D-21-36898The Effects of Microbiome-Targeted Therapy on Cognitive Impairment and Postoperative Cognitive Dysfunction - A Systematic ReviewPLOS ONE

Dear Dr. Kinjo,

Thank you for submitting your manuscript to PLOS ONE. After careful consideration, we feel that it has merit but does not fully meet PLOS ONE’s publication criteria as it currently stands. Therefore, we invite you to submit a revised version of the manuscript that addresses the points raised during the review process.

We look forward to receiving your revised manuscript.

Kind regards,

Alessandro Putzu, M.D.

Academic Editor

PLOS ONE

Journal Requirements:

Additional Editor Comments:

A number of issues have been identified in the review process. While we feel that this manuscript shows promise, we also think that a major revision is needed. Before we can make a final decision about this manuscript we want to offer you the opportunity to revise and resubmit the manuscript.

The study is original and interesting. I think that the methods section should be improved (reporting and clarity).

I have some comments:

1- Methods. It is unclear if the study followed a protocol and if the methodology was conceived before study start. Did you register the protocol before study start? The lack of a registered protocol is a major limitation that should be discussed.

2- Methods. The study included “aged” patients. Did you follow any cut-off? Any age limit?

3- Methods. How did you manage missing outcome data? Did you contact corresponding authors?

4- Methods. Supplemental details on risk of bias assessment should be reported in the supplementary material (e.g.: what other bias include? How did you finally rate a study? etc.)

5- Results. References of major exclusions should be included in the supplementary material.

6- Results. Much more details on risk of bias assessment should be reported in the supplementary material (full details on each item). The authors reported that all items of each study had low risk of bias; this is possible but very improbable.

7- Results. Overall risk of bias evaluations for each trial should be reported.

8- Discussion. The certainty of evidence supporting microbiome-target therapy is very low. This therapy that should be tested in large, randomized trials.

9- Discussion. A paragraph on study’s limitations should be included.

Minor comments:

1- Methods. I suggest to include a PRISMA 2020 checklist.

2- Results. A PRISMA 2020 flow-chart should be used.

Reviewers' comments:

Reviewer's Responses to Questions

**Comments to the Author**

1. Is the manuscript technically sound, and do the data support the conclusions?

Reviewer #1: Yes

Reviewer #2: Yes

Reviewer #3: Yes

2. Has the statistical analysis been performed appropriately and rigorously? 

Reviewer #1: N/A

Reviewer #2: N/A

Reviewer #3: N/A

3. Have the authors made all data underlying the findings in their manuscript fully available?

Reviewer #1: Yes

Reviewer #2: Yes

Reviewer #3: Yes

4. Is the manuscript presented in an intelligible fashion and written in standard English?

Reviewer #1: Yes

Reviewer #2: Yes

Reviewer #3: Yes

5. Review Comments to the Author

Reviewer #1: The review is timely and full of novelty. The authors summarized the recent progress in the role of gut microbiota in PND and FMT used for prevention and treatment of PND. Although large number of studies showed that gut-brain axis plays a critical role in neuropsychiatric diseases, there is little evidence showing FMT for PND treatment. Several points should be addressed before accpetance.

1 As the authors maintained, neuroinflammation may be an important factor of PND. The question is whether neuroinflammation is caused by abnormal gut microbiota, or surgery-induced peripheral inflammation?

2 We know that FMT will not see significant changes in the short term. At least, the regulation of brain function by FMT is a chronic process. Is it better to treat before or after surgery?

Reviewer #2: Thank you for the opportunity to review the manuscript “The Effects of Microbiome-Targeted Therapy on Cognitive Impairment and Postoperative Cognitive Dysfunction - A Systematic Review” from Kinjo et al. In this systematic review authors tried to answer the question: if microbiome targeted therapy can improve postoperative cognitive dysfunction(POCD)?

It is now known that Gut microbiota is linked to many illness and increasing research is focused on defining causal link and potential therapeutic applications. The idea is novel and important because currently we don’t have any therapies to prevent POCD. Overall writing is clear but methods/discussion can be improved.

Methods: should be more specific in the abstract. Was this systematic review registered? Readers will benefit from separate Exposure and outcomes section in the method. which probiotics/ how they are usually administered etc. A table showing risk/benefit/available commercial product [for human or experimental use], of MMT will be useful. This will highlight differences between various interventions.

Discussion: While the authors summarized the current research findings. They should discuss, based on the research, which specific MMT regimen shows promise and should be investigated further. The goal is to guide readers and researcher, what next? How these MMT should be studied?

Authors did write “further basic and clinical research, and more randomized controlled studies are needed that correlate specific probiotic and prebiotic formulas with measurable biological and clinical outcomes relevant to cognitive impairment” Some specifics should be added rather than a generalized statement.

Assessment of outcome, POCD is complicated by varied resource intensive methods, and should be discuss further. Also, how much improvement in POCD can result from MMT should be discussed. Given perioperative POCD pathophysiology is not fully understood.

“Please correct word spacing in tables”. The main results seem to be copy pasted. Please be consistent and specific, tell what improved and by how much (if possible) and was it significant?

Reviewer #3: In this manuscript Sugita and co-workers summarize the extant knowledge on prebiotic and probiotic therapy for cognitive impairment and POCD from published studies in humans and animals. Overall I think this article makes a valuable contribution and it is thoughtfully organized and well written. I have minor suggestions by section below:

Abstract:

1) The conclusion does not really convey much information other than a positive outlook on the use of microbiome interventions. The authors should either shorten it, or better yet offer some more sophisticated interpretation of their results.

Introduction:

1) Given that many of the studies reviewed here are conducted in animals, the authors should discuss the similarities and differences in human and animal model microbiota and they should give a clear understanding of the limitations and benefits of studies in each.

2) The authors should include a paragraph on what has been published in human and animal models about the changes in the microbiota that are seen in the perioperative setting due to surgery, anesthesia, oxygen, opiates, and other known causes of dysbiosis. It is important to set up for the reader why it is likely that disruption of the microbiota are a cause of POCD.

Methods

1) A reference should be given for the Cochrane risk and bias tool

2) It would be helpful and provide useful validation for this work if the authors cited other published studies that use similar literature review methodologies.

Results

1) It would be helpful if the authors came up with a diagram that summarized the various mechanisms by which dysbiosis is thought to alter cognitive outcomes that also includes some indication of how the interventions might have helped.

Discussion

1) It would be great if the authors could include a paragraph suggesting where the gaps are in the literature and what should be done next to address them. This is a really helpful feature of many reviews of this kind.

6. PLOS authors have the option to publish the peer review history of their article (what does this mean?). If published, this will include your full peer review and any attached files.

Reviewer #1: **Yes: **Chun Yang

Reviewer #2: **Yes: **Kamal Maheshwari

Reviewer #3: No

---

## [Author Response · Author response to Decision Letter 0]

8 May 2022

Please see the attached rebuttal letter.

---

## [Decision Letter · Decision Letter 1]

15 Jun 2022

PONE-D-21-36898R1

The Effects of Microbiome-Targeted Therapy on Cognitive Impairment and Postoperative Cognitive Dysfunction - A Systematic Review

PLOS ONE

Dear Dr. Kinjo,

Thank you for submitting your manuscript to PLOS ONE. After careful consideration, we feel that it has merit but does not fully meet PLOS ONE’s publication criteria as it currently stands. Therefore, we invite you to submit a revised version of the manuscript that addresses the points raised during the review process.

We look forward to receiving your revised manuscript.

Kind regards,

Alessandro Putzu, M.D.

Academic Editor

PLOS ONE

Journal Requirements:

Additional Editor Comments:

Thank you for your great work on the manuscript. I still have few comments.

1-Abstract. I suggest to report that you included animal and human studies.

2-Results. SYRCLE risk of bias results should be reported. Full details on each study should be reported in the supplementary material.

3-Results. The structure of the Results section is unclear and confusing in my opinion. Results should include systematic review results, an objective assessment of the evidence. Statements on potential effects or interpretation of the results shuld be moved to the discussion section.

I suggest to report results of systematic review in the Results section (“Gut-microbiota and MTTs on Cognitive Function in Rodents” and “Gut-microbiota and MTTs on Cognitive Function in Humans”).

In my opinion the paragraph “Neuroinflammation, aging, and POCD” and “Microbiome-targeted therapy (MTT)” may be moved in the discussion.

4-Supplementary material. No need to include the PROSPERO protocol in the supplementary material; it is online and freely available. Please remove it.

5-Supplementary material. The PRISMA 2020 checklist for the abstract is missing.

6-Supplementary material. The ‘Microbiome Exclusion-Final’ table should include information on major exclusions. Some more information allowing the retrieval of each manuscript should be reported (e.g.: first author + year of publication + doi; first author + year of publication + journal info; full reference according to journal style). The actual form (first author + year of publication) is not informative enough (the manuscripts could not be retrieved).

Reviewers' comments:

Reviewer's Responses to Questions

**Comments to the Author**

1. If the authors have adequately addressed your comments raised in a previous round of review and you feel that this manuscript is now acceptable for publication, you may indicate that here to bypass the “Comments to the Author” section, enter your conflict of interest statement in the “Confidential to Editor” section, and submit your "Accept" recommendation.

Reviewer #1: All comments have been addressed

Reviewer #3: All comments have been addressed

2. Is the manuscript technically sound, and do the data support the conclusions?

Reviewer #1: Yes

Reviewer #3: Yes

3. Has the statistical analysis been performed appropriately and rigorously? 

Reviewer #1: N/A

Reviewer #3: N/A

4. Have the authors made all data underlying the findings in their manuscript fully available?

Reviewer #1: Yes

Reviewer #3: Yes

5. Is the manuscript presented in an intelligible fashion and written in standard English?

Reviewer #1: Yes

Reviewer #3: Yes

6. Review Comments to the Author

Reviewer #1: All the comments have been well addressed, and that the review will attract widely attentions and bring increasing interets in the topic.

Reviewer #3: In this article, Kinjo and colleagues conduct a systematic review to test the hypothesis based on published data that probiotics can improve cognitive outcomes after surgery. The review uses both preclinical and clinical data, which was an excellent choice in my opinion given that there is relatively little of either and that it opens up the opportunity to consider potential mechanisms. Overall, I am in favor of publication of this article, which I think will be a valuable contribution to the literature. I have a few brief comments below:

1. The introduction is considerably undercited. There is a wealth of literature on the gut and other microbiota and surgical outcomes that should be cited in a more complete fashion. Also, the authors note that anesthetics, analgesics, and antibiotics have effects on the microbiota but do not have any citations to back this assertion.

2. There are several other reviews of probiotics and surgical outcomes. It’s important to frame the introduction to distinguish how this manuscript is a valuable addition to the literature.

3. While systematic reviews of this kind are less common in animal models than in clinical studies and the practices for them are less clear, they should still attempt to conform to the better recognized approach for clinical studies. I would argue that the preclinical approach should also have been registered/deposited as well.

4. What is gray literature?

5. Can the authors provide some citations validating their methodology for assessing bias? Particularly as it relates to animal studies

6. The vast majority of studies were excluded due to lack of relevance. This does suggest the possibility that the search terms were not well designed for the study.

7. PLOS authors have the option to publish the peer review history of their article (what does this mean?). If published, this will include your full peer review and any attached files.

Reviewer #1: **Yes: **Chun Yang

Reviewer #3: No

---

## [Author Response · Author response to Decision Letter 1]

30 Jul 2022

Please see attached letter for our responses.

---

## [Decision Letter · Decision Letter 2]

8 Sep 2022

PONE-D-21-36898R2The Effects of Microbiome-Targeted Therapy on Cognitive Impairment and Postoperative Cognitive Dysfunction - A Systematic ReviewPLOS ONE

Dear Dr. Kinjo,

Thank you for submitting your manuscript to PLOS ONE. After careful consideration, we feel that it has merit but does not fully meet PLOS ONE’s publication criteria as it currently stands. Therefore, we invite you to submit a revised version of the manuscript that addresses the points raised during the review process.

Reviewer 1:

The review is timely and full of novelty. The authors summarized the recent progress in the role of gut microbiota in PND and FMT used for prevention and treatment of PND. Although large number of studies showed that gut-brain axis plays a critical role in neuropsychiatric diseases, there is little evidence showing FMT for PND treatment. Several points should be addressed before acceptance.

1 As the authors maintained, neuroinflammation may be an important factor of PND. The question is whether neuroinflammation is caused by abnormal gut microbiota, or surgery-induced peripheral inflammation?

2 We know that FMT will not see significant changes in the short term. At least, the regulation of brain function by FMT is a chronic process. Is it better to treat before or after surgery?

Reviewer2:

In this manuscript Sugita and co-workers summarize the extant knowledge on prebiotic and probiotic therapy for cognitive impairment and POCD from published studies in humans and animals. Overall I think this article makes a valuable contribution and it is thoughtfully organized and well written. I have minor suggestions by section below:

Abstract:

1) The conclusion does not really convey much information other than a positive outlook on the use of microbiome interventions. The authors should either shorten it, or better yet offer some more sophisticated interpretation of their results.

Introduction:

1) Given that many of the studies reviewed here are conducted in animals, the authors should discuss the similarities and differences in human and animal model microbiota and they should give a clear understanding of the limitations and benefits of studies in each.

2) The authors should include a paragraph on what has been published in human and animal models about the changes in the microbiota that are seen in the perioperative setting due to surgery, anesthesia, oxygen, opiates, and other known causes of dysbiosis. It is important to set up for the reader why it is likely that disruption of the microbiota are a cause of POCD.

Methods

1) A reference should be given for the Cochrane risk and bias tool

2) It would be helpful and provide useful validation for this work if the authors cited other published studies that use similar literature review methodologies.

Results

1) It would be helpful if the authors came up with a diagram that summarized the various mechanisms by which dysbiosis is thought to alter cognitive outcomes that also includes some indication of how the interventions might have helped.

Discussion

1) It would be great if the authors could include a paragraph suggesting where the gaps are in the literature and what should be done next to address them. This is a really helpful feature of many reviews of this kind.

We look forward to receiving your revised manuscript.

Kind regards,

Chan Chen

Academic Editor

PLOS ONE

Journal Requirements:

Reviewers' comments:

Reviewer's Responses to Questions

**Comments to the Author**

1. If the authors have adequately addressed your comments raised in a previous round of review and you feel that this manuscript is now acceptable for publication, you may indicate that here to bypass the “Comments to the Author” section, enter your conflict of interest statement in the “Confidential to Editor” section, and submit your "Accept" recommendation.

Reviewer #1: All comments have been addressed

Reviewer #3: All comments have been addressed

2. Is the manuscript technically sound, and do the data support the conclusions?

Reviewer #1: Yes

Reviewer #3: Yes

3. Has the statistical analysis been performed appropriately and rigorously? 

Reviewer #1: Yes

Reviewer #3: N/A

4. Have the authors made all data underlying the findings in their manuscript fully available?

Reviewer #1: Yes

Reviewer #3: Yes

5. Is the manuscript presented in an intelligible fashion and written in standard English?

Reviewer #1: Yes

Reviewer #3: Yes

6. Review Comments to the Author

Reviewer #1: This is a timely, well organized, and well written review. All the questions have been addressed. The mansucript could attract attentions.

Reviewer #3: Thank you for the opportunity to review this manuscript. Comments have been addressed, I agree with publication.

7. PLOS authors have the option to publish the peer review history of their article (what does this mean?). If published, this will include your full peer review and any attached files.

Reviewer #1: **Yes: **Chun Yang

Reviewer #3: No

---

## [Author Response · Author response to Decision Letter 2]

10 Jan 2023

As the editorial office suggested, previous version of the manuscript has been attached.

---

## [Editor Report · Decision Letter 3]

16 Jan 2023

The Effects of Microbiome-Targeted Therapy on Cognitive Impairment and Postoperative Cognitive Dysfunction - A Systematic Review

PONE-D-21-36898R3

Dear Dr. Kinjo,

We’re pleased to inform you that your manuscript has been judged scientifically suitable for publication and will be formally accepted for publication once it meets all outstanding technical requirements.

Kind regards,

Emily Chenette

Editor in Chief

PLOS ONE
---

## [Editor Report · Acceptance letter]

23 Jan 2023

PONE-D-21-36898R3 

The Effects of Microbiome-Targeted Therapy on Cognitive Impairment and Postoperative Cognitive Dysfunction - A Systematic Review 

Dear Dr. Kinjo:

I'm pleased to inform you that your manuscript has been deemed suitable for publication in PLOS ONE. Congratulations! Your manuscript is now with our production department. 

Kind regards, 

on behalf of

Dr Emily Chenette 

Staff Editor

PLOS ONE